# Obesogenic Programming Effects during Lactation: A Narrative Review and Conceptual Model Focusing on Underlying Mechanisms and Promising Future Research Avenues

**DOI:** 10.3390/nu13020299

**Published:** 2021-01-21

**Authors:** Junilla K. Larsen, Lars Bode

**Affiliations:** 1Behavioural Science Institute, Radboud University, PO Box 9104, 6500 HE Nijmegen, The Netherlands; 2Department of Pediatrics and Larsson-Rosenquist-Foundation Mother-Milk-Infant Center of Research Excellence, University of California, San Diego, CA 92101, USA; lbode@health.ucsd.edu

**Keywords:** lactation, obesogenic programming, infant weight outcomes, human milk composites, microbiota, flavor exposure

## Abstract

Animal studies have consistently demonstrated that maternal obesity and a high-fat diet during lactation enhances obesity risk in the offspring. However, less is known about these potential obesogenic programming effects in obese humans. We propose three important pathways that may explain obesogenic programming effects of human breastmilk. First, human milk components and hormones may directly affect child eating and satiety characteristics. Second, human milk constituents can affect child microbiota that, in turn, may influence child eating and weight outcomes. Third, human milk composition may affect child eating and weight outcomes through flavor exposure. We reviewed a few very recent findings from well-powered longitudinal or experimental human research with regard to these three pathways. Moreover, we provide a research agenda for future intervention research with the overarching aim to prevent excessive pediatric weight gain during lactation and beyond. The ideas presented in this paper may represent important “black box” constructs that explain obesogenic programming effects during lactation. It should be noted, however, that given the scarcity of studies, findings should be seen as working hypotheses to further test in future research.

## 1. Introduction

Animal studies have consistently demonstrated that maternal obesity and high-fat diets during lactation enhance obesity risk in the offspring [1,2,3,4]. However, less is known about these potential obesogenic programming effects in obese humans. We propose three important pathways that may explain obesogenic programming effects of human breastmilk. First, human milk components and hormones may directly affect child eating and satiety characteristics. Second, human milk constituents can affect child microbiota that, in turn, may influence child eating and weight outcomes. Third, human milk composition may affect child eating and weight outcomes through flavor exposure. We shortly reviewed recent findings from well-powered longitudinal or experimental human research with regard to these three pathways. Moreover, we provide a research agenda for future intervention research based on current knowledge gaps with the overarching aim to prevent excessive pediatric weight gain. It is important to note that this paper does not provide a systematic review of the literature. Instead, we provide a simple narrative review that addresses the pathways in the proposed conceptual model. We highlight distinctions, potential areas of overlap, and lacking areas in the different study designs and literatures. Illustrative examples are included in the interest of stimulating research.

## 2. Maternal Adiposity, Human Milk Composition, and Infant Weight Outcomes

Erikson published a review in 2018 proposing that the association of human milk composition with infant growth was still speculative and needed further investigation, as the included studies were limited in their methodological design [5]. In 2019, Isganaitis and colleagues published the first comprehensive longitudinal analysis of the human milk metabolome in relation to both maternal and infant obesity. Although this study raised the possibility that some milk constituents might play a pathogenic role in mother-to-child transmission of obesity, the sample size was relatively small [6]. Longitudinal, well-designed studies using gold standard methodologies and sufficient statistical power are highly needed to enhance knowledge on this topic [5], but are obviously still limited in terms of causal inferences. Three larger sampled longitudinal studies have been published since then [7,8,9]. These studies show evidence for elevated levels of certain milk macronutrient constituents, including oligosaccharides, monosaccharides (e.g., mannose), and sugar alcohols (e.g., lyxitol) in the milk of obese/overweight women, associated with the development of infant adiposity over time [7,9]. In addition, positive associations with child weight outcomes (i.e., weight gain) have been found in recent smaller sampled studies when specific fatty acid components (e.g., Omega-6/Omega-3 ratio) were considered [10,11], that may be of importance, given the findings of a recent systematic review that these lipid components were higher in women with excessive body weight [12].

In addition to macronutrients, elevated concentrations of leptin and insulin in human milk may also play an important role [8]. Insulin and leptin are actively transported from the mother to the infant through human milk more consistently from the beginning on [13,14]. Notably, while leptin concentrations in human milk were positively associated with lower child weight outcomes shortly after birth (i.e., first weeks to months) [15,16], positive associations of leptin with accelerated increases in child weight outcomes have been found later on [16]. Leptin is a well-known satiety hormone that suppresses appetite, which might explain why exogenous leptin from human milk may initially be associated with lower infant weight. Though speculating, leptin might play a role in later accelerated increases in child weight outcomes through leptin resistance, reinforcing “reward eating” beyond caloric requirements [17]. To conclude, human milk composition may play a pathogenic role in mother-to-child transmission of obesity through child eating and satiety characteristics [18]. However, mediation by child gut microbiota and flavor exposure may also play a role (see Figure 1).

### 2.1. Gut Microbiota

Gut microbiota are established during the first three years of life [19]. It has been theorized that associations between human milk composition of obese mothers and child weight outcomes may be mediated by child gut microbiota [20]. As an illustrative example, Pekmez and colleagues recently found that certain breastmilk components (e.g., human milk oligosaccharide (HMO) diversity and Lacto-N-neotetraose (LNnT)) were lower in breastmilk of infants with excessive growth and that those breastmilk components were subsequently positively associated with fecal branched short-chain fatty acid (SCFA) concentrations, which may explain the rapid weight gain of these children [21]. This is an exciting future research avenue to pursue. Although gut microbiota are linked to obesity risk [22,23], better research designs, including standardization of techniques to classify gut microbiota structure and function [23], well-powered studies, and longitudinal and experimental research designs are highly needed to better understand cause and effect.

### 2.2. Flavor Exposure

Based on a narrative review of the literature in 2017, Ventura suggested that breastfeeding and human milk flavors may also shape food preference links to obesity [24]. Theoretically, flavor exposure during lactation aligns with known key components of preference development (i.e., repeated exposure, variety exposure, and associative conditioning) [25] that may explain why breastfeeding may shape these healthy food preference links that might prevent obesogenic eating patterns [24]. To date, a recent systematic review reports only two randomized controlled trials (RCTs) in total that have examined causal effects of exposure to specific flavors during lactation on flavor acceptance [26]. Both RCTs performed by the group of Mennella and colleagues focused on learning to like vegetables and found that repeated ingestion of certain “healthy” flavors increased infants’ acceptance of these flavors in cereal several months later. Moreover, the impact of flavor exposure was specific to the exposed flavor and did not generalize to a novel unexposed flavor [26]. Specifically, in one of these RCTs, Mennella and colleagues found that one month of flavor exposure during lactation was more effective when it began at two weeks postpartum compared to later periods, suggesting that timing is important in this repeated vegetable flavor exposure during lactation [27]. Future studies should further examine whether and how increased acceptance of vegetable flavors may prevent later obesogenic eating patterns.

## 3. Conclusions

Much needs still to be learned about how human milk composition of obese women can be changed to prevent unhealthy mother-child obesity programming during the lactation period. Our review may offer guidance on a couple of mechanisms that need to be further examined, as previously mentioned. Moreover, we propose a future intervention research agenda that may act on human milk components and/or child satiety and eating characteristics during lactation, thereby potentially preventing excessive pediatric weight gain during lactation and beyond. Nevertheless, we need to be careful with research examining whether and how breastfeeding among mothers who are obese may affect mother-to-child transmission of obesity, as it may stigmatize obese women (e.g., “your milk is not good enough”) and demotivate lactation intentions among “new” mothers who are obese, while it is known that these women already have more problems with lactation to begin with [28,29,30]. Breastfeeding has several positive effects beyond child weight outcomes [31,32], and it is therefore important to seek for ways in which we can optimize breastfeeding among “obese women”, while simultaneously preventing potential “obesity programming” during lactation.

## 4. Future Intervention Research Agenda

We will discuss two promising lines of intervention research directly acting on postulated human milk components (i.e., maternal stress and lifestyle, see Figure 1). We will explain why we think that these are important topics to intervene on during lactation to prevent childhood obesity. Moreover, we suggest a third line of intervention research (i.e., early weight-related parenting) that is a bit outside of the scope of this piece but is nevertheless deemed important as it may influence child satiety and eating characteristics during lactation. We will provide a couple of future research avenues bridging these general lines of intervention research. 

A first important intervention topic that may directly impact human milk composition is maternal stress. Stress is defined as: “a negative emotional experience accompanied by predictable biochemical, physiological, cognitive, and behavioral changes that are directed either toward altering the stressful event or accommodating to its effects” [33]. Maternal stress during and after pregnancy is very common [34] and interventions that reduce stress have been shown to reduce weight and improve obesity-related eating behaviors [35]. Maternal stress probably plays a role in the development and maintenance of obesity through mechanisms of cognition, lifestyle behaviors, physiology, and biochemistry [36]. Maternal stress may also be linked to child obesity [37,38,39], but the mechanisms explaining this link are largely unknown. We propose two main pathways that may explain this link during lactation (see Figure 1), that is, one physiological pathway (i.e., through human milk composition) [40,41] and one behavioral one (i.e., through maternal lifestyle and weight-related parenting) [42,43] affecting child weight-related behaviors [44]. Future research may examine whether and how intervention on maternal stress may impact human milk components and child weight outcomes through lactation-specific pathways (i.e., child satiety and eating characteristics, child gut microbiota, and flavor exposure).

A second important intervention topic that may directly impact human milk composition and subsequent childhood obesity is maternal lifestyle (e.g., nutrition, physical activity, and smoking) [45,46]. Specifically, physical activity is known to decline during pregnancy and the postnatal period [47] and, in turn, maternal weight retention or weight gain after birth is very common, being a key contributor to both maternal obesity [48,49] and rapid weight gain trajectories in children [50]. Notably, animal studies suggest that maternal lifestyle during lactation is a stronger driving factor on impact of offspring body weight than maternal obesity alone [4,51]. Moreover, a recent animal study shows beneficial effects of maternal exercise on mouse offspring’s weight and other metabolic outcomes, causally mediated by milk oligosaccharides (i.e., 3′-SL), with a similar exercise-induced increase in 3′-SL observed in humans [52]. While this is an exciting area of research suggesting direct effects of maternal lifestyle (i.e., exercise) on human milk composition with consequences for infant health, future lifestyle preventive intervention efforts among humans are needed that assess lactation-specific mechanisms, including for instance effects of human milk oligosaccharides on child gut microbiota and child weight outcomes. As an illustrative example of potential mediation through infant gut microbiota, a scoping review found that postnatal maternal smoking was significantly associated with specific infant gut microbiota that, subsequently, were linked to childhood obesity [53]. Although causality cannot be inferred, and effects on lactation and human milk composition were not assessed, these findings are important and warrant further research.

To date, previous lifestyle multi-component (e.g., diet and activity) intervention studies have shown beneficial effects on maternal lifestyle and weight outcomes during pregnancy and beyond [54,55], but most studies neither reported on infant’s weight status, nor assessed (lactation-specific) intervention mediators [56]. Moreover, effects on maternal weight outcomes were often small and established in efficacy trials with biased samples [57]. To prevent childhood obesity, there is a need of more effective and implementable maternal lifestyle interventions [57] that start during the preconception period [58], use a socioecological framework and theory-informed intervention design [59], include well-known effective core components and strategies for behavioral change [60], and use a flexible person-centered rather than a one-size-fits-all approach [61]. Maternal lifestyle interventions may also pay more attention to automatic processes underlying lifestyle behaviors [62]. We recently developed a process framework that guides the “personalized” selection of different intervention strategies targeting automatic processes to reduce unhealthy lifestyle behaviors [63].

A third important intervention topic that does not impact human milk composition, but may importantly impact child satiety, eating, and weight characteristics during lactation is responsive feeding. Evidence from a systematic review of intervention studies suggests that providing responsive feeding guidance from early lactation stages on to mothers to recognize child hunger and satiety cues is linked to “normal” child weight status development, thus preventing excessive weight gain [64]. Moreover, a couple of universal parenting prevention programs targeting early feeding and positive parenting skills specifically during the prenatal/infancy developmental stage show significant effects on infant weight-related or dietary outcomes [56]. As such, weight-related positive and responsive parenting interventions prove promising in the infancy and lactation period [65]. However, most parenting programs have been aimed at later developmental stages. More “early” weight-related parenting interventions targeting diverse populations and obesity risk behaviors beyond diet and physical activity are needed [66].

In addition to targeting these three intervention topics as separate factors, future research should also bridge a gap between the literature that considers how maternal stress, lifestyle/obesity, and weight-related parenting affect child weight outcomes. We propose that intervention efforts combining maternal stress and lifestyle or maternal stress and parenting are particularly important in preventing excessive weight gain in both mothers and children (see Figure 1 dotted lines), by attention to automatic lifestyle or parenting aspects [67]. To date, there are a couple of promising intervention studies that combine stress and lifestyle intervention components among mothers in the period around pregnancy [68,69], but these studies neither reported child weight outcomes, nor assessed (lactation-specific) intervention mediators. Future studies may examine whether and why combined intervention on maternal stress and lifestyle during (pregnancy and) lactation impacts child weight outcomes through previously mentioned lactation-specific mechanisms. Moreover, very few studies have examined interventions combining stress and parenting, but those that did so suggest that these kind of interventions are a potential way to attenuate the risk of childhood obesity in very young children [70]. Responsive parenting interventions that act on child eating and satiety mechanisms might yield stronger effects in combination with intervention on maternal stress (also through lifestyle) and human milk composition. Future intervention trials should further examine interaction effects between different intervention components (i.e., stress versus parenting) explaining pediatric weight gain, while simultaneously assessing (lactation-specific) relevant mechanisms of action. Finally, beyond lactation-specific mechanisms, many pathways tie maternal stress to child obesity, but research is needed that ties stress to lifestyle to parenting to child obesity in a single study.

In conclusion, evidence remains inconclusive due to the limited number of published data. To date, the three lactation-specific pathways within the conceptual model are weakly supported, since they are grounded on a very short reflective review of a few recent studies instead of a comprehensive systematic analysis of the literature with formal quality assessment per paper. Moreover, the factors suggested for a “future research agenda”, such as maternal stress and lifestyle, and early weight-related parenting, are not yet supported by intervention studies demonstrating accurately that these factors affect lactation and human milk composition among humans. Thus, more research is needed, with several randomized controlled intervention studies already underway [71,72]. Finally, the exemplar intervention strategies are not meant to be exhaustive. For instance, administering antibiotics also has the potential to affect gut microbes through breastfeeding, possibly influencing childhood obesity [73]. Nevertheless, this paper highlights constructs and processes that may have utility for bridging diverse intervention areas in the lactation period to prevent excessive pediatric weight gain. While the focus of the intervention targets lays on mothers, we propose that use of a socioecological framework and intervention design [59] may yield more promising effects. That is, intervention efforts should pay attention to broader social networks around mothers and to community-based implementable approaches [57]. Moreover, our model focuses on the period of lactation. However, intervention efforts will probably be more effective when started early on, during pregnancy or even during the preconception period [58], with continuing intervention efforts beyond lactation. The ideas presented in this paper may represent important “black box” constructs that explain obesogenic programming effects during lactation. It should be noted, however, that given the scarcity of studies, findings should be seen as working hypotheses to further test in future randomized controlled trials using an “experimental medicine” approach, examining basic mechanistic processes as part of intervention trials [74].

## Figures and Tables

**Figure 1 nutrients-13-00299-f001:**
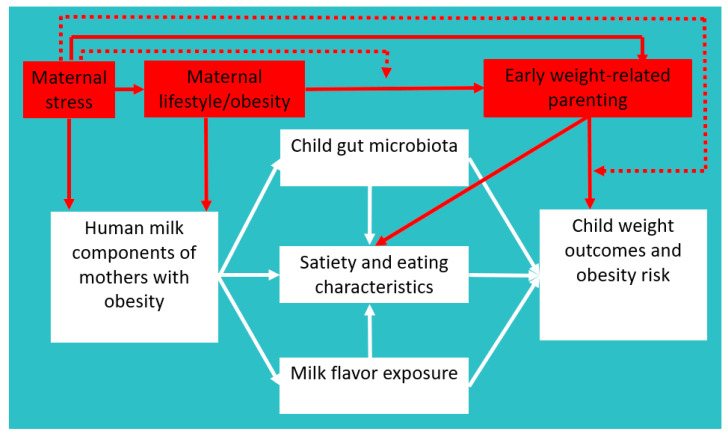
A theoretical model explaining the link between human milk composition among mothers with obesity and child weight outcomes. The red parts indicate exciting future research avenues; dotted lines reflect moderation. The exemplar intervention strategies are not meant to be exhaustive.

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
