# Peer review of "Obesogenic Programming Effects during Lactation: A Narrative Review and Conceptual Model Focusing on Underlying Mechanisms and Promising Future Research Avenues"

_nutrients, 2021, doi:10.3390/nu13020299_

Round 1
Reviewer 1 Report
The manuscript by Larsen and Bode on “Obesogenic Programming Effects during Lactation: A Position Paper on Underlying Mechanisms and Promising Future Research Avenues” is well written. It contains two interesting parts a) on the association of specific human milk components with anthropometry of mother and infants and b) on potentially promising interventions how to decrease the obesity risk in offspring of obese mothers.
By itself each part is interesting to read and seems to help in the planning of further studies. Nevertheless, these two parts of the manuscript seem not closely related. The examples mentioned, which show associations of maternal mood and behavior with human milk components are not fully aligned with the examples mentioned in the first part.
One would expect that the suggested future research is based more specifically on demonstrated or postulated mechanisms, which are related to human milk components. Thus, I think the manuscript requires a middle part linking the mechanism and research suggestion parts.
Reviewer 2 Report
Based on the title alone, this manuscript seems to promise addressing an interesting subject: Obesogenic programming effects during lactation, and the underlying mechanisms. However, the manuscript provides a narrative review with superficial data and its structured is a confusing way:
A position paper aims to support healthcare providers’ decision-making when the amount and/or quality of available evidence prevent to develop a clinical practice guideline (Ingravallo 2014). It should represent more than the opinion of the authors but should present opinions supported by scientific societies, professional organizations, or government agencies (Ingravallo 2014, Bala 2018). A position paper should contain a strong background information and discussion providing a complete understanding of the issue involved, and the rationale behind the position set forth (Ingravallo 2014). Hence, a position statement should be supported at least by a systematic review (Gough 2020). This manuscript is too ambitious to be a Position paper, since it provides a simple narrative review without a formal quality assessment (Grant 2009), and only reflects the opinion of the authors.
The Authors propose three pathways that may explain obesogenic programming effects of human breastmilk. Twenty-four longitudinal or experimental studies in humans were selected and were superficially described, without a formal quality assessment.
The proposed “Future Research Agenda” is not in line with the aforementioned three pathways, since it is focused on maternal stress, lifestyle/obesity, and weight-related parenting, and the mechanisms proposed by the Authors to be investigated are not directly related with lactation, thus not supporting the pathways presented in Figure 1.
References
Bala M, Kashuk J, Moore EE, et al.. Establishing position papers by the WSES. World J Emerg Surg. 2018 Jan 15;13:1. doi: 10.1186/s13017-018-0163-8.
Gough D, Davies P, Jamtvedt G, et al. Evidence Synthesis International (ESI): Position Statement. Syst Rev. 2020 Jul 10;9(1):155. doi: 10.1186/s13643-020-01415-5.
Grant MJ, Booth A. A typology of reviews: an analysis of 14 review types and associated methodologies. Health Info Libr J. 2009 Jun;26(2):91-108. doi: 10.1111/j.1471-1842.2009.00848.x.
Ingravallo F, Dietrich CF, Gilja OH, Piscaglia F. Guidelines, clinical practice recommendations, position papers and consensus statements: definition, preparation, role and application. Ultraschall Med. 2014 Oct;35(5):395-9. English, German. doi: 10.1055/s-0034-1385158.
Round 2
Reviewer 1 Report
I would like to thank the authors for considering/discussing the Points raised
Author Response
We would like to thank the reviewer for re-reviewing this piece and for his or her remark that the points have been considered/discussed.
Reviewer 2 Report
Changes made to the manuscript improved some aspects, but main weaknesses remain:
- This is a narrative review and this should be specified in the title “A Narrative Review Focusing…”
- The conceptual model proposed by the authors is weakly supported, since it is grounded on a very short reflective review instead of a comprehensive systematic analysis of the literature with formal quality assessment per paper.
- To support obesogenic programming effects during lactation, only 6 studies were cited in which three mechanisms have been studied, related with: 1) nutrients, insulin and leptin n human milk (references 7-9); 2) child gut microbiota related with human milk composition of obese mothers (reference 17 and 18); and 3) healthy food preference acquired through breastfeeding, potentially preventing obesogenic eating patterns (reference 21).
- The factors suggested for a “Future Research Agenda”, such as maternal stress and lifestyle, and early weight-related parenting, are not supported by studies demonstrating accurately that these factors affect lactation. Therefore, a conceptual model including these factors as pathways have very limited interest, since they are mainly speculative, based on reflective assumptions instead of an evidence based rationale.
